# Border Tourism Development Strategies in Kaleybar Compared to Regional Rivals

Amin Safdari Molan [1,*], Ebrahim Farhadi [1], Lucia Saganeiti [2] and Beniamino Murgante [3]

1 Faculty of Geography, University of Tehran, Tehran 1417935840, Iran; e.farhadi71@ut.ac.ir
2 Department of Civil, Building-Architecture and Environmental Engineering, University of L'Aquila, Via G. Gronchi, 18, 67100 L'Aquila, Italy; lucia.saganeiti@gmail.com
3 School of Engineering, University of Basilicata, 85100 Potenza, Italy; beniamino.murgante@unibas.it
* Correspondence: a_safdari@ut.ac.ir

**Abstract:** A richness of tourism attractions has given Iran global importance within its border. Iran is a country with a huge cultural heritage, and is rich in historical monuments from different eras. The variety and diversity of cultural symbols allows tourists traveling in Iran to experience the cultures of other countries. The border areas of the country are therefore becoming increasingly attractive for tourism due to their distinctive social, economic and political position and the presence of many historical and natural attractions. This study analyzes border tourism in Kaleybar city using a descriptive–analytical method with a Meta-SWOT model (new strategic planning tool), with the final goal of economic development and the improvement of the welfare of the people. Through a literature review, the current and future capabilities and challenges of the county of Kaleybar as a border city are examined, and future goals and ways to achieve them have been developed using the opinions of experts and scholars via the Delphi technique. For this purpose, the Meta-SWOT model has been used. Meta-SWOT is based on resource-based theory (RBV). Data collection has been done several times using the opinions of 39 experts. After analyzing the conditions and recognizing and determining the capacities and skills of Kaleybar and its regional rivals, the results of the research show that the most important advantage of Kaleybar city concerns the existence of many attractions of a natural character. A higher strategic suitability is also ensured by the presence of parks and coastal sidewalks that attract important internal and foreign investments in this region. On the other hand, the component of political and governmental factors in attracting foreign tourists has the highest effective power, and the component of attention to integrated management in the field of tourism in the country has the highest degree of urgency.

**Keywords:** border tourism; strategic spatial plan; meta-SWOT; kaleybar county; regional rivals





## 1. Introduction

Tourism is a social, cultural and economic phenomenon that entails the movement of people to countries or places outside their usual environment for personal or business/professional purposes [1]. Tourism is an important factor of economic systems and social life in many countries of the world, and therefore the countries that recognize their tourism potentials can use it as a source of capital enhancement. Tourism is primarily a leisure activity, along with other activities [2]. It is necessary to utilize all the country's capacities and capabilities to create sustainable and comprehensive development to replace the older, oil-based sources of income with new ones. In this regard, the development of the tourism industry, which economists regard as the third-most dynamic and quickly growing economic phenomenon after the oil and automotive industries, is considered as a basic need of the country [3]. Therefore, it is necessary to study the obstacles to developing this industry in the different regions of the country [4]. Iran, in terms of tourism attractions, has very good climate diversity; in terms of cultural, natural and social heritage, it is comparable with developed countries in the tourism industry. The country has great potential

for developing its tourism industry; as a matter of fact, in terms of monuments, it is among the top nine countries, and in terms of ecotourism attractions, it is among the top 10 countries [5]. However, some evidence indicates a number of challenges and weaknesses in the Iranian tourism industry. An examination of the Iranian tourism industry statute reveals that, in terms of the tourism sector's share of GDP, Iran ranks 86th out of 174 countries in the world, and is third among the Persian Gulf States after UAE and Qatar. Besides this, in terms of investment in the tourism industry, Iran ranks 172 out of 174 countries in the world and is last in the Middle East. In addition, in terms of tourism industry value in 2005, Iran was ranked 43rd out of 174 countries worldwide [6]. The richness of tourism attractions in Iran has made it a global environment within a border. The variety of attractions in Iran is so great that it creates enough motivation to travel to this country for almost any taste, because Iran is a country with a lot of historical monuments and cultural heritage, left over from different eras. Besides this, cultural symbols in Iran are so diverse and various it is as if a tourist traveling inside Iran goes from one country to another country ([7–9], pp. 13–31).

The border regions of Iran are of great importance in terms of tourism due to their special social, economic and political position, and the existence of many very important historical and natural heritage sites. Therefore, in recent years, the weaknesses, opportunities and threats of the tourism sector have been focused on by Iranian researchers and planners in order to exploit the tourism potential and opportunities that the country offers (for more about historical tourism in the Azerbaijan region, look to [8], pp. 63–77). Kaleybar is one of Iran's border counties, and has great historical and natural heritage potential, which is very important because it is located in the Free Trade Zone area. Additionally, along the communication route and the common border between the three neighboring countries, namely Azerbaijan and Armenia, and with Turkey, there is a direct connection through the city of Urmia, where trade and economic transactions take place, and on the other hand, this helps to develop tourism and introduce attractions and potential. Through neighboring other European countries, the region has strategic advantages. It is located in a very powerful place in terms of natural and pristine resources and historical monuments that are useful for the development of natural, historical and archaeological, recreational, medical, and health tourism in this city and surrounding cities ([10–12]; for more about historical tourism in the Azerbaijan region, look to [8], p. 78). The county Kaleybar was selected as a biosphere reserve by the United Nations many years ago due to the forests of Azerbaijan and its pristine and rich nature. In 1978, UNESCO registered the Arasbaran Protected Area as a biosphere reserve in the list of most valuable natural sites in the world. This region is currently one of the nine biosphere reserves (genetic ponds) in the country [13]. Considering the young population of the county, the development of tourist infrastructure in Kaleybar can boost employment and bring prosperity. As it has young people and an active labor force (average age between 22 and 49 in the city of Kaleybar), this can lead to the development of tourism, the labor market and various activities [14]. This article investigates the potentials and capabilities of the Kaleybar frontier county, and compares them with Kaleybar's neighbors. Besides this, a strategic map of the county devised via Meta-SWOT software is provided.

## 2. Literature Review

### 2.1. The Concept of Tourism

Tourism refers to the complex of events and experiences during the journey of a tourist. This process also includes activities such as planning a trip, traveling to a destination, staying, returning, and even reminiscing about it [15,16]. Today, one of the most promising activities is related to the development path [17]. Tourism was one of the most developed industries in the mid-twentieth century, and has been the key to development and economic growth ([18], p. 63). The tourism industry is one of the most significant aspects of the approach to evaluating a nation, because it can simultaneously display the best of one nation's heritage in a variety of economic, social, environmental, cultural, political, and

technological contexts ([19], p. 168) [20,21]. Tourism, as a significant form of human activity, can have major impacts. These impacts are very visible in the destination region, where tourists interact with the local environment, economy, culture and society. Hence, it is common to consider the impacts of tourism under the headings of socio-cultural, economic and environmental impacts. This convention is followed in the three chapters that follow this introduction to tourism's impacts. In fact, tourism issues are generally multi-faceted, often having a combination of economic, social and environmental dimensions. Therefore, when considering each of the types of impact in turn, it should be remembered that the impacts are multi-faceted, often problematic, and not as easily compartmentalized as is often portrayed. In other words, tourism impacts cannot easily be categorized as solely social, environmental or economic, but tend to have several inter-related dimensions. It should also be noted that much tourism planning and management is related to tourism's impacts in destinations and resorts. The impacts of tourism can be positive or beneficial, but also negative or detrimental [22–24]. Tourism's impacts are likely to change over time as a destination area develops [21]. According to Wall [22], key factors contributing to the nature of the impacts include the type of tourism activities engaged in, the characteristics of the host community in the destination region, and the nature of the interaction between the visitors and residents. Davison [24] suggested a range of similar influences and also included the importance of time and location in relation to tourism impacts. In stressing the importance of the "where" and the "when", Davison [24] claimed these influences set tourism's impacts apart from those of other industrial sectors. In relation to tourism being concentrated in space, Davison indicated that tourism production and consumption, unlike many other industrial activities, takes place in the same location. This means that the tourist consumes the product in the tourist destination. Therefore, tourism's impacts are largely spatially concentrated in the tourism destination. The World Tourism Organization's (WTO) Global Code of Ethics for Tourism (United Nations World Tourism Organization [25]) was adopted at the thirteenth WTO General Assembly in Santiago, Chile, in 1999, and later ratified by the UN General Assembly in 2001. It represents two years of effort on the part of the UNWTO to develop a set of guidelines that would, in the words of Francesco Frangialli, then UNWTO Secretary-General, act as "a frame of reference for the responsible and sustainable development of world tourism". Therefore, in this research, the negative and positive effects of tourism are considered, seeking to decrease the negative effects by facilitating the development of sustainable border tourism in the city of Kaleybar. The development of sustainable border tourism in the city of Kaleybar has been structured by planning the development of tourism by means of strategic planning via the analytical and strategic model of Meta-SWOT.

*2.2. Tourism Planning*

Simmons [26] says that as tourism has risen in significance as a tool for regional development, it must also face the call for increasing public scrutiny and involvement. Therefore, tourism planning is fundamental for regional development. Planning is accepted as a critical approach for future development guidance. Without planning the development process, it will be accidental, vulnerable, and likely to fail. Today, tourism planning is pursued in many countries and regions that seek to promote controlled tourism [27]. Obviously, when tourism is part of management activities, management strategies and frameworks need to make sure that tourism activities preserve and support natural and socio-cultural values [28] (See also [29,30]). In regional tourism planning, the capacities and capabilities of tourism destinations must be identified to determine the position of each destination in the development process of the region. Identifying this position can develop the reasonable expectations and rational treatment of tourism destinations in the context of planning and investing in accordance with its resources and capacities, as well as the allocation of facilities ([26,31,32], p. 43). Theorists and planners try to use different perspectives and identify advantages and limitations, in order to achieve the success of tourism destinations in response to the demands of tourists on the one hand, and to

improve residential areas on the other hand [31,33]. Understanding the tourist attractions of each region and providing executive strategies to improve the tourism situation will help develop tourism and economic prosperity in the region.

### 2.3. Strategic Planning in the Tourism Industry

Tourism is one of the most important sectors of the country's industry and economy, and it has become increasingly important with globalization. Tourism, with the possibility of preserving values, beliefs and ancient traditions, can be an expression of the values of societies at the national and local levels. Urban tourism, by creating employment opportunities and providing income for local residents, as well as encouraging infrastructure development, provides the possibility of sustainable development and urban development [34–36]. Today, tourism is one of the main sources of income for developed countries and some developing countries. Meanwhile, the top countries in the tourism industry, each according to the existing potentials in the country, have been able to provide the basic factors for the development of tourism. According to the forecasts of world economic experts, the tourism industry has been one of the most profitable industries in the world from 2000 to 2020, even surpassing the oil and automobile industries, and is considered as one of the three major industries in the world ([37,38], p. 563). Strategic management and planning are the best and most important tools for controlling this type of change or conscious presence in the market and increasing the possibility of organizational innovation ([35,38], p. 563). On the other hand, in each organization, the strategies, frameworks of programs, policies, activities, and decisions of managers and employees relate to achieving the goals and objectives of the organization. Therefore, research related to strategy development, evaluation, and the selection of the best approach for an organization in order to strengthen it while reducing weaknesses and threats is urgent ([39], p. 98).

### 2.4. Tourism and the Development of Local Communities

Tourism is recognized as a strategy for local economic development. Businesses that are involved in tourism can create jobs, bring new money into the local economy, and bring diversity into the foundations of the local economy. Due to the reduction in long-term extraction and production resources, economic diversity is essential for long-term development in marginal and rural areas ([40], p. 20). Components of tourism include tourists, those with different businesses related to the industry, tourism managers and the host community (local people). All these need to benefit from tourism in order for this industry to succeed in a sustainable way. Tourists look to make the most of the destination at the lowest cost, while businesses are looking to maximize profits (in the short term). Host communities are interested in long-term earnings and employment as net profits from the industry. Acquaah [41] stated that in order to achieve sustainability in tourism, all those involved in the industry (government, private sector, NGOs and local people) must be fully connected and participate in all parts of its development ([42], p. 14). Tourism as a service has this feature: the place of production and consumption is the same as the destination of tourism. However, this destination is not designed for tourists, and is first and foremost a place where people live [43–45]. The World Tourism Organization's ethical principles emphasize that local communities should be involved in tourism activities and equally benefit from cultural, social, and economic benefits, especially in the case of direct and indirect jobs ([46], p. 289), [47,48]. However, many studies have shown that local communities in third-world countries may benefit little from tourism because they have little control over how the industry develops. They are not able to compete in terms of available capital resources against foreign investors, and their attitudes and opinions are rarely heard ([49], p. 88). The only types of local participation that have the ability to change existing patterns of power and unequal development are those formed within the local community ([50], p. 240). Local communities play an essential role in the development of tourism, as they are very important in providing suitable situations for tourists. The role of local communities in influencing tourism development has become clearer. McIntyre argues

that local communities need to organize themselves at all levels to play a more effective role in development, as well as in communication with the government. Kepe argues that local communities should play a very active role to ensure the positive benefits of tourism. Local communities need to work extensively with non-governmental organizations to teach others in the community to consider tourism development projects ([51], p. 155). Without empowerment at the local and national levels, national efforts to develop tourism in practice will fail ([52], p. 100). In order to develop tourism, it is suggested that empowerment be considered as a multidimensional process, in order to provide a consultative process with specific characteristics (such as the ability to make decisions, and the capacity to implement and use these decisions, to accept responsibility for decisions, actions and their consequences, etc.) that can provide for communities ([53], p. 112). It is widely believed that a participatory development approach facilitates the implementation of the principles of sustainable development by delegating control and management to the local community, and enforcing decision-making based on consensus and the equal flow of interests to all those affected by development ([52], p. 1440). Finally, it can be said that tourism has a significant impact on local communities for tourism purposes. The industry can be an important source of income and employment for local people, and it can also pose a threat to the social environment of a region, as well as its cultural and natural heritage. However, if properly planned and managed, it can be considered as a force to protect them ([54,55], pp. 8–12).

*2.5. Sustainable Tourism Development*

Sustainable tourism development is defined as follows: "Sustainable tourism development is a process that meets the needs of current tourists and the host community while supporting and strengthening future opportunities." In fact, in this definition, movement leads to the management of all resources in a way that meets economic, social, aesthetic and ecological needs while maintaining integrity in terms of energy, water, air, habitat, wildlife, ecology, and biodiversity [3], (pp. 160–163), [56,57], p. 301, [58], p. 459.

In the world of development literature, the sustainable development paradigm has been considered by experts since the 1970s. Earlier attention to sustainable tourism began in the 1960s, with the identification of the potential effects of mass tourism and the impact of tourism activities on the economy, environment and culture of tourist spots ([58,59], p. 326). The uncontrollable growth of mass tourism has destroyed natural, social and cultural resources, such as the destruction of cultural heritage and local identity, increased crime rates, overcrowding, and other environmental issues in the host regions [3,59,60]. Sustainable development is a type of development strategy that manages all natural and human assets and resources, such as financial and physical resources, to increase wealth in the long term. Sustainable development is opposed to any policy or practice that somehow endangers the interests of future generations ([61,62] pp. 288–304).

## 3. Methodology

We follow a pragmatic qualitative research approach. According to Patton, methodological appropriateness is the primary criterion for assessing methodological quality in pragmatic research [63,64].

Meta-SWOT was first introduced in 2012 by a team of three consisting of Agarwal, an assistant professor of computer science at Norbert College in the United States, and two colleagues at the Department of Commercial Management at the same university. This approach is inspired by resource-based theory.

In this theory, it is assumed that successful organizations, cities and villages are created due to their exclusive capabilities. As a result, a firm's resources available for determining strategic activity are more critical than its external environment. Meta-SWOT views works based on the creation and development of resources with the aim of influencing the surrounding environment, and instead of being passive in the face of the external environment, it encourages the organization to influence it. In the present study, we have

tried to evaluate the study area based on a theory of the value, scarcity and irreplaceability of resources and internal capabilities of RIO (V) compared to rivals. It should not be forgotten that no factor is the weak point except in relation to competition. Meanwhile, the value criterion is not evaluated in the process of reviewing resources and capabilities unless the resources and capabilities are consistent with the external environment.

This is because, by definition, a resource or capability is valuable when it has the ability to make optimal use of opportunities and neutralize threats to the external environment. In fact, planners cannot judge the value of existing resources or capabilities apart from assessing the external environment. They suggest the concept of strategic fit. All calculations in the Meta-SWOT software are based on the average score of the data ([65], pp. 13–15).

Strategic planning is used as a tool to better perform activities and achieve the desired goals in the future, in line with set goals. Strategic planning can also be used as a response to the problems in the tourism industry.

The Meta-SWOT model runs in a program consisting of a title window and seven interconnected windows. Its purpose is to guide decision-makers through an integrated process from the initial stage of the brainstorm to creating a ranked list of strategic priorities. This tool enables unlimited reviews of inputs, as decision-makers change their assessment during a planning activity.

Meta-SWOT is based on resource-based theory (RBV). This theory asserts that unique resources and capabilities are the main factors of continuous competitive advantage.

The method of the present research is descriptive–analytical, and its purpose is applied. The method of data collection and the analysis of the required information is documentation and surveying. The Meta-SWOT analytical technique has been used to formulate the development strategy and explain the goals, resources, capabilities and environmental factors. Data collection has been carried out several times using the opinions of 39 experts. As such, except for the objectives of the research, all stages of the research were identified and prioritized with the help of experts. The research findings show that Meta-SWOT can eliminate many of the shortcomings and inadequacies of SWOT by avoiding subjective decisions and using a systematic and high-precision technique. On the other hand, this model tries to show that sustainable development and competition can be operational when tourism areas make the best use of internal resources and capabilities by rejecting the exogenous development perspective. Meta-SWOT can identify the position of Kaleybar and its villages in comparison with rivals in different competitive dimensions, determine the value, scarcity, imitation and irreplaceability of its resources and capabilities, examine the impact of internal factors on controlling threats or optimally using external opportunities, and assess the constructive roles of goals, resources and capabilities in the optimal use of the competitive advantages of sustainable tourism in rural areas. In this way, we can enable development and prosperity in the border city of Kaleybar and its villages.

*Study Area*

Kaleybar city, the center of Kaleybar county with an area of 2071.97 km$^2$ and an average height of 1180 m above sea level, is located in the northeastern region of the East Azerbaijan province (see Figure 1). The center of Kaleybar county is Kaleybar city, which is 165 km away from the center of the province. This city is bounded on the south by the cities of Ahar and Vaezaghan, on the west by Jolfa, on the east by the Mughan plain, and on the north by the Qarabağ in Azerbaijan. (see Appendix A: some of tourism potential and touristic places of the Kaleybar county provided). Kaleybar county is located between 38°27′ N, 49°27′ E. Its population in the 2011 census was 48,837 people [66].

The study area has great potential for developing tourism. Moreover, Kaleybar city is a border city, which is located in connection with three countries that have very similar cultural and linguistic roots and customs (Azerbaijan, Iran, Armenia and also sometimes Turkey) as well as natural tourist attractions and historical cities (see Appendix A). However, in spite of these potentials, due to the neglect of these capabilities given a lack of

knowledge and planning and not paying attention to these areas in the development of the country, Kaleybar city has also suffered from economic poverty. Therefore, with the development of border tourism in Kaleybar city via a strategic planning approach in the Meta-SWOT program environment, it is possible to provide guidance for decision-makers, and in this way, it can enable development and prosperity in the border city of Kaleybar and its villages.

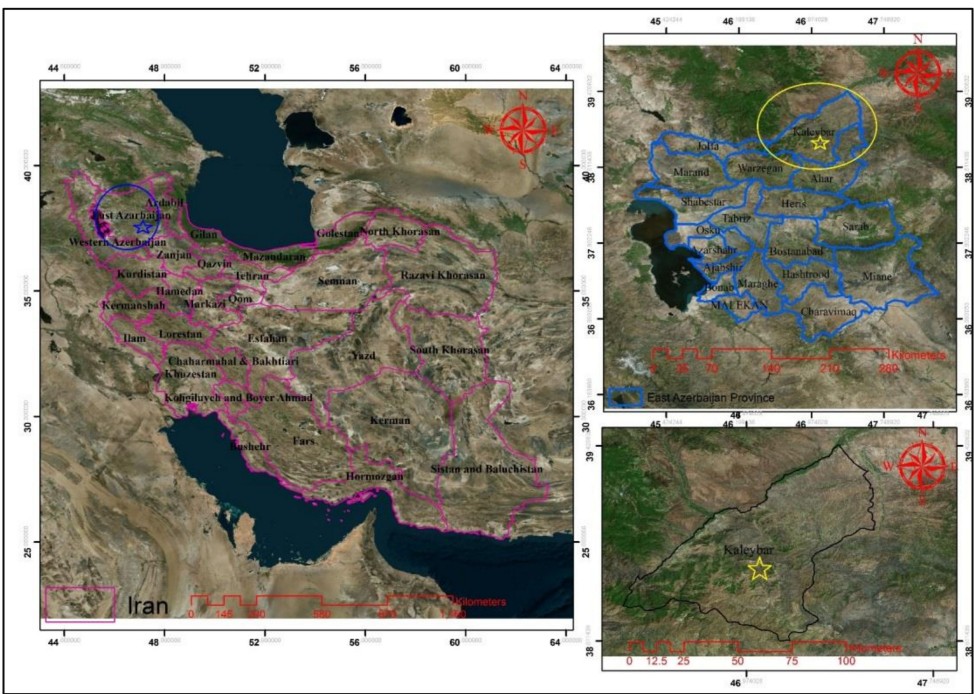

**Figure 1.** Drawing by the authors, 2021.

## 4. Discussion and Results

### 4.1. Data Analysis

#### 4.1.1. Identify Research Objectives

In the first step, the main objectives of the research were identified ([55], p. 119, [45], p. 278). The main purpose of this research was to determine the position of Kaleybar county in the border tourist area. In this regard, goals had to be set. Due to the fact that these goals do not have the same weight, experts were asked to prioritize these goals according to their importance at three levels (high, medium and low). Table 1 shows the objectives of the study along with their importance. After compiling this table and determining the priorities, this information was entered into the software.

**Table 1.** Objectives set for border tourism in Kaleybar.

| Purpose | Priority Level |
|---|---|
| Investigation of border areas' capacities in the tourism field | High |
| Comparative study of tourism development in border counties | High |
| Investigating the condition of the residents of the region in the tourism field | Low |
| Containing infrastructures of tourism activities' development | Medium |

Source: Authors' findings, 2021.

The next step was to identify the tourist resources and capabilities of Kaleybar (see Appendix A). After studying the library and documentary resources, including the strategic plan of tourism in Kaleybar, as well as using the opinion of experts, this information was derived. At this stage, efforts have been made to identify resources and capabilities in order to achieve the desired goals ([67], p. 1021). Because these resources and capabilities do not

have the same weight, their effectiveness has been measured based on the percentage of impact they could have on achieving the research goals (Table 2):

**Table 2.** Border tourism resources and capabilities in Kaleybar county.

| Row | Sources and Capacity | Percentage |
|---|---|---|
| 1 | Existence of hotels and resting places | 8 |
| 2 | Existence of abundant natural attractions | 10 |
| 3 | Proper and favorable weather | 9 |
| 4 | Existence of valuable cultural and historical works throughout the county | 10 |
| 5 | Containing beach sidewalks and parks | 8 |
| 6 | Proper spatial quality (healthy environment) | 11 |
| 7 | Development of basic facilities and infrastructure | 12 |
| 8 | The possibility of attracting large domestic and foreign investments to the region | 13 |
| 9 | Extensive cultural and economic ties between the people on both sides of the border | 12 |
| 10 | Including a lot of traditional customs and crafts and interesting music | 7 |

Source: Authors' findings, 2021.

It was then necessary to classify the main factors of success (resources and capabilities) into two competitive dimensions, according to the resources and capabilities extracted in the previous step. Additionally, some factors in the selected dimensions were removed from the list, due to their incompatibility, in order to achieve more logical results. Due to the fact that the nature of tourism here is based on competition, the rivals who are competing with Kaleybar in terms of their role and position in the region had to be selected. Therefore, Parsabad, Germi, Jolfa, Maku, Bazarghan and Khoda afarin counties were determined as rivals of Kaleybar (Figure 2).

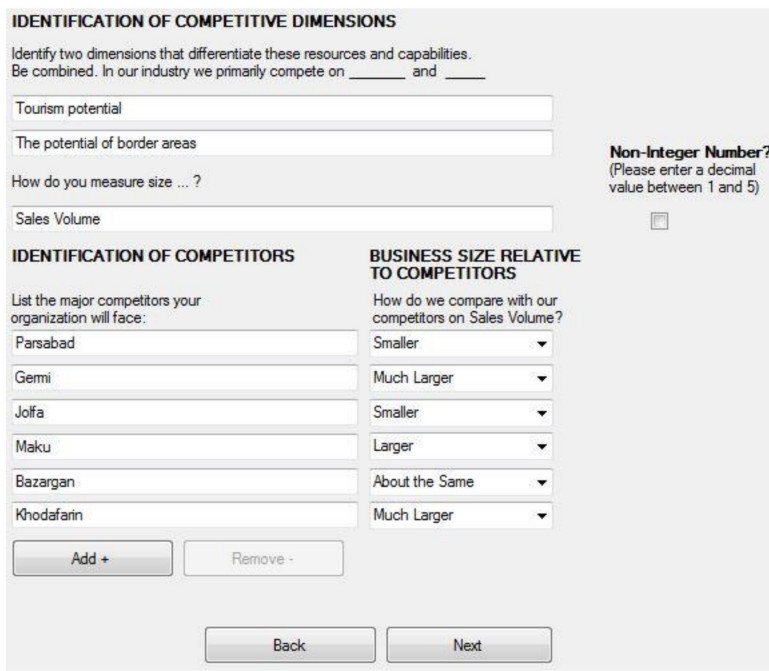

**Figure 2.** Identification of competitive dimensions and rival cities of Kaleybar.

We then had to measure the border tourism capabilities of Kaleybar compared to rival counties. At this stage, the county under study was compared with its rivals based on the capabilities and resources identified, ranging from very high to very low in a range of five. Table 3 shows Kaleybar's position based on well-known capabilities compared to its rivals (Table 3).

**Table 3.** Comparison of Kaleybar with its regional rivals in terms of available resources and capabilities.

| Sources and Capacity | Pars-Abad | Germi | Jolfa | Maku | Bazargan | Khodaafarin |
|---|---|---|---|---|---|---|
| Existence of hotels and resting places | Equal | Better | Worse | Better | Worse | Much Better |
| Existence of abundant natural attractions | Better | Better | Equal | Equal | Better | Equal |
| Proper and favorable weather | Better | Better | Equal | Equal | Better | Equal |
| Existence of valuable cultural and historical works throughout the county | Better | Better | Worse | Much better | Better | Better |
| Containing beach sidewalks and parks | Worse | Equal | Worse | Equal | Equal | Worse |
| Proper spatial quality (healthy environment) | Better | Much better | Equal | Better | Better | Better |
| Development of basic facilities and infrastructure | Worse | Equal | Much worse | Worse | Worse | Worse |
| The possibility of attracting large domestic and foreign investments to the region | Worse | Equal | Worse | Worse | Worse | Equal |
| Extensive cultural and economic ties between the people on both sides of the border | Worse | Worse | Worse | equal | Worse | Equal |
| Including a lot of traditional customs and crafts and interesting music | Better | Better | Better | Much better | Much better | Better |

In this step, according to the comparison made in the previous steps and the obtained results, the following diagram was drawn. Thus, Jolfa has been identified as the biggest rival in the region to Kaleybar in terms of tourism capacities, with a weight of 4.26; Bazargan with a weight of 4.11, Pars Abad with a weight of 4, Maku with 3.34, Khoda afarin with a weight of 3.44 and Germi with a weight of 3.26 are the other rivals of Kaleybar (Figure 3). See also the Table 4, which shows weight of competitive advantage of each rival area.

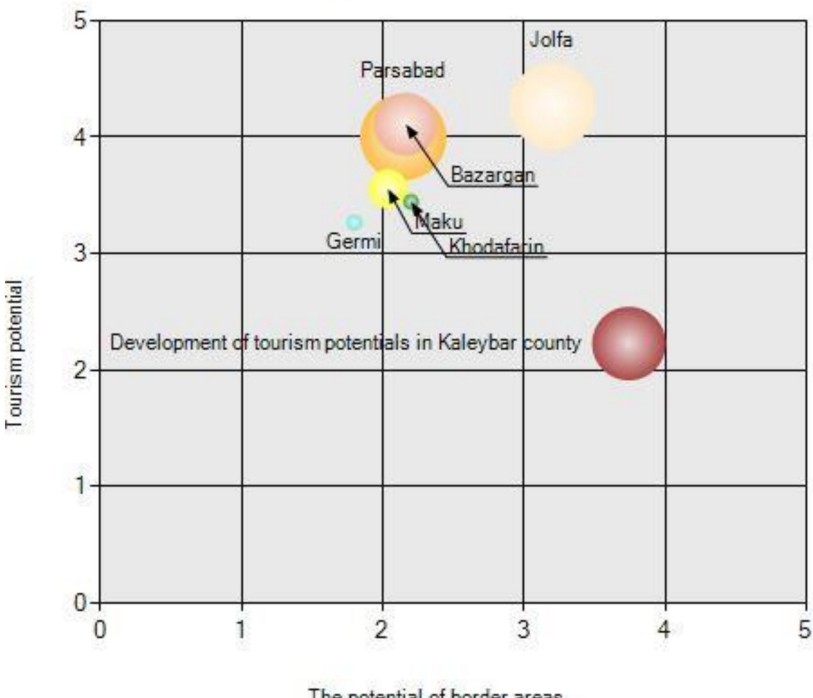

**Figure 3.** Competitive map of Kaleybar compared to rivals in border areas.

**Table 4.** Weight of competitive advantage of each rival area.

| County | Tourism Capacity | | The Potential of Border Areas | | Total | | Score | |
|---|---|---|---|---|---|---|---|---|
| | Absolute | Normalized | Absolute | Normalized | Absolute | Normalized | Absolute | Normalized |
| Pars Abad | 2/14 | 0/95 | 4 | 1/05 | 6/14 | 2/01 | 4 | 4 |
| Germi | 1/8 | 0/79 | 3/26 | 0/86 | 5/06 | 1/66 | 1 | 1 |
| jolfa | 3/2 | 1/41 | 4/26 | 1/13 | 7/46 | 2/54 | 6 | 6 |
| Macu | 2/03 | 0/90 | 3/55 | 0/94 | 5/59 | 1/84 | 2 | 2 |
| Barzarhan | 2/16 | 0/95 | 4/11 | 1/08 | 6/27 | 2/04 | 5 | 5 |
| Khoda afarin | 2/2 | 0/97 | 3/44 | 0/91 | 5/64 | 1/88 | 3 | 3 |

Source: Authors' findings, 2020.

At this stage, via resource-based theory, the resources and capabilities were measured according to scarcity, imitation and irreplaceability ((V) RIO) on a scale of five (strongly agree, agree, neutral, disagree, strongly disagree). Here, the criterion of being valuable was not measured, because the value of resources and capabilities can be measured by comparison to external factors. The experts' responses to the terms of the agreement are as follows: Rareness (R)—Our rivals cannot do that; Imitability (I)—Our rivals are not able to imitate this feature; Irreplaceable (O)—Do we use this factor as our compensation policy? Table 5 indicates imitability and irreplaceable for the resources and capabilities measured according to scarcity, imitation and irreplaceability.

**Table 5.** Imitability and irreplaceable.

| Sources and Capacity | Scarcity | Imitability | Irreplaceable |
|---|---|---|---|
| Existence of hotels and resting places | Disagree | Strongly agree | Neutral |
| Existence of abundant natural attractions | Strongly agree | Strongly disagree | Agree |
| Proper and favorable weather | Strongly agree | Strongly disagree | Agree |
| Existence of valuable cultural and historical works throughout the county | Agree | Disagree | Agree |
| Containing beach sidewalks and parks | Agree | Disagree | Agree |
| Proper spatial quality (healthy environment) | Neutral | Agree | Disagree |
| Development of basic facilities and infrastructure | Disagree | Agree | Strongly disagree |
| The possibility of attracting large domestic and foreign investments to the region | Neutral | Agree | Strongly disagree |
| Extensive cultural and economic ties between the people on both sides of the border | Neutral | Agree | Disagree |
| Including a lot of traditional customs and crafts and interesting music | Agree | Neutral | Neutral |

Source: Research findings, 2021.

### 4.1.2. Identify PESTEL Related Factors

At this stage, we identified external factors such as political, economic, socio-cultural, technological, ecological and legal (PESTEL) affecting the development of border tourism in the study area. This analysis is effective in identifying the potential opportunities as well as the potential risks of expanding each set ([68], p. 2). Generally, this analysis deals with macro-environmental assessment and current time assessment ([69], p. 335), because macro-factors have the ability to make fundamental changes to the environment and thus the collection. The PESTEL framework is based on political, economic, social, technological, environmental and legal factors ([70], p. 16). In this step, the external factors, degree of stagnation and dynamism, degree of effectiveness, probability of increase and degree of stability of these factors were evaluated. Table 6 shows effective environmental factors out of the region's control.

**Table 6.** Effective environmental factors out of the region's control.

| Priority | Increase Probability | Effectiveness | Weight | Factor |
|---|---|---|---|---|
| High | High | High | High | Paying attention to integrated management in country tourism |
| High | High | Medium | Medium | Existence of rival-free zones in the region, outside Iran's borders |
| High | High | High | High | Lack of cooperation between qualified centers in the region for the development of tourism facilities |
| Weak | Low | Weak | Weak | The Ghareh Bagh crisis as a regional neighbor |
| High | Medium | Medium | Medium | Existence of specialized tourism forces in areas outside the borders of this region |
| High | Medium | Medium | Medium | Increasing facilities and services in rival recreational areas and attracting more tourists |
| High | High | High | High | Political and governmental factors in attracting foreign tourists |
| High | High | High | High | Lack of a suitable situation for attracting foreign investors |

Source: Authors' findings, 2021.

Weighting (static and dynamic): How important are these factors compared to average future challenges?

Interpretation: What is the effect of this factor on future success?

Probability of increase: What is the probability of an increase in this factor during the planning period?

Extraordinary degree: To what extent is this factor required?

*4.2. Fitness Strategy*

4.2.1. Proportions of Resources, Capabilities, and External Factors

At this stage, experts were asked to evaluate the resources and capabilities assigned to external factors. Strategic appropriateness was calculated by matching internal factors with external factors. In general, in strategic planning, there are two distinct internal and external types of factors. Public studies emphasize the influence of both internal and external factors in the planning environment [41,71].

4.2.2. Proportions of Sources, Capabilities, and Research Objectives

At this stage, experts were asked to evaluate the impacts of resources and capabilities on research objectives. Strategic fit was calculated by matching internal factors with goals. Here we looked at how resources and capabilities support the research goals.

4.2.3. Strategic Map or Strategic Spatial Plan

At this stage, the strategic plan was drawn. In the strategic plan, external resources, capabilities, and factors are analyzed based on three criteria: (A) Proximity of resources and capabilities with external factors. (B) Horizontal and upper side of factors. (C) The size of the bubbles. In the strategic plan, internal resources and capabilities are denoted by the blue bubble (Figure 4). The horizontal axis (X) indicates imitation and organizational appropriateness, and the vertical axis (Y) indicates the degree of strategic fit of the factors. The research findings indicate that the resources and capabilities of border tourism development in Kaleybar city have different weights. The components of hotels and places to rest (X: 0/799. Y: 3. Hubble Size: 4/5), the existence of abundant natural attractions (X: 1.Y: 3. Hubble Size: 4/5), optimality of weather (X: 0/899. Y: 3. Hubble Size: 1/5), valuable cultural and historical monuments in the city (X: 1. Y: 1. Hubble Size: 3), parks and coastal walkways (X: 0/799. Y: 3. Hubble Size: 0/799), proper spatial quality (healthy environment) (X: 0/99. Y: 3. Hubble Size: 0), development of basic facilities and infrastructure (X: 0/83. Y: 3. Hubble Size: 0), ability to attract large internal and foreign investment into the region (X: 1/03. Y: 3. Hubble Size: 0), extensive cultural and economic connections between people on both sides of the border (X: 1/08. Y: 3. Hubble Size: 0), large number of traditional customs and handicrafts and attractive traditional music (X: 0/69. Y: 3. Hubble Size: 0) show

the status of the tourism resources and capabilities of Kaleybar. External factors are also marked by orange bubbles. The horizontal axis indicates the strength of the factors and the vertical axis and bubble size indicate the degree of necessity of the factors. The components of attention to integrated management in the field of tourism in the country (X: 5. Y: 1. Hubble Size: 5), existence of rival-free zones in the regions that are Iran's borders (X: 1/79. Y: 2. Hubble Size: 2), lack of cooperation between qualified centers in the development of tourism facilities in the region (X: 3/20. Y: 1. Hubble Size: 5), the Ghareh Bagh crisis as a regional neighbor (X: 0/80. Y: 4. Hubble Size: 1), existence of tourism specialists in areas outside the borders of this region (X: 1/79. Y: 3. Hubble Size: 3), increasing facilities and services in competing recreational areas and attracting more tourists (X: 1/79. Y: 3. Hubble Size: 3), political and governmental factors in attracting foreign tourists (X: 4. Y: 1. Hubble Size: 0), and lack of suitable grounds for attracting foreign investors (X: 3/2. Y: 2. Hubble Size: 5) indicate the impact of macro-environmental factors. Meanwhile, the closeness of two factors indicates that those internal factors can support external factors.

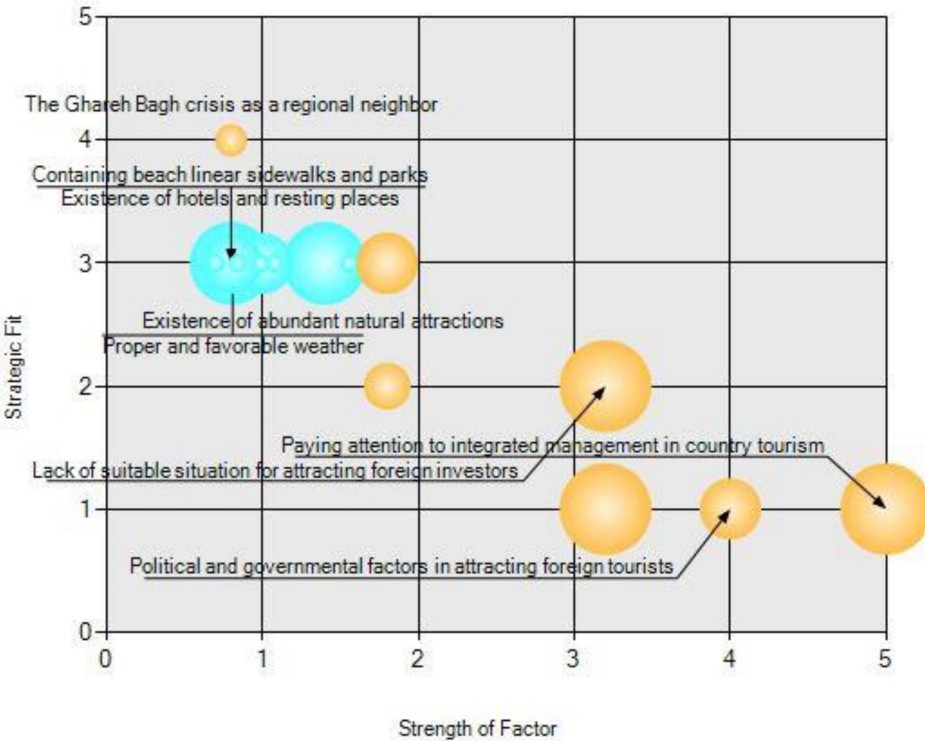

**Figure 4.** Map of border tourism development strategies in Kaleybar city.

The findings of this study are consistent with other tourism studies in other aspects, which confirm that the research can be intensified, as the results indicate that the link between culture and tourism is the most obvious aspect of the contribution of culture to local development. In addition, a growing number of economists now accept that there is a third form of "capital" or "economic good", which is crucial for the proper functioning of the economic system of production, consumption and global welfare. This distinct category consists of the endowment of natural resources and environment in an economy, which refers to natural heritage [72]. There is a strong consensus in the literature that tangible–intangible, movable–immovable and spiritual heritage assets create competitive advantages and innovation, and become promoters of the regeneration and growth of a destination [73]. Culture, nature, cultural and natural heritage, and development have been making an incalculable contribution toward improving human livelihood and well-being in lasting and sustainable ways [74]. Tourism can influence the social and cultural

cohesion of the community's residents [75]. The various branches of the economy are interdependent, with certain mutual influences as a result of the exchange that takes place between them. Increases in tourism activity produce a number of positive effects on other sectors of the economy. Specialists consider three elements that contribute to measuring the economic impact of tourism expenditure: direct, indirect and induced effects [76,77]. Direct effects refer to expenditures in the tourism sector, based on a list of specific tourism products developed by the WTO and the OECD. Therefore, when tourists spend money on hotels, restaurants, transport, communications and retail outlets, this will give rise to direct revenue, government revenue, employment, and the direct imports of goods and services. Indirect effects refer to intermediate consumption for the production of goods and services in the tourism sector. These are goods and services that travel companies buy from their suppliers, forming the supply chain of the tourism sector. Indirect effects can harness the production of local products, and are important for encouraging the acquisition of goods and services produced locally in the tourism sector in order to maximize the economic impact of the income generated by tourism in a country or region. Induced contribution measures the GDP and jobs supported by the spending of those employed directly or indirectly in the tourism industry. It measures spending on food and drink, leisure, transport, housing and household goods, etc. The innovative aspect of this study is the use of the Meta-SWOT strategic analytical model, and also its innovation in the field of sustainable border tourism, which may be very effective in the local dimension of tourism development and in the border cities of different countries. As mentioned, we have also confirmed the research results.

## 5. Conclusions

Iran's border areas have a high potential for tourism. Most of these areas have many tourist attractions. Kaleybar plays a special role in the history of tourism development in the northeastern region of the East Azerbaijan province; the existence of pristine natural places and historical sites in this region has led to the creation of tourism capabilities in this county. Kaleybar can be a source of income for the city and the region, and can reduce poverty and unemployment in the region. However, the fact is that the tourism industry has not been very successful in this area. Most developed countries, despite having macroeconomic resources and income, prefer to invest in the tourism industry. The direct presence of visitors and tourists in a border area, in addition to economic development and cultural exchanges, makes that part of the country a safe place for living and tourism. Therefore, in this study, an attempt has been made to evaluate the resources and capabilities of Kaleybar county using Meta-SWOT software. Meta-SWOT software, by rejecting Brunza's development perspective, reveals that sustainable development and competition in an area become operational when said area makes good use of its unique resources and capabilities, and then sets its desired goals within a specific time horizon.

Strategic planning is used as a tool to better carry out activities and achieve desired goals in the future. Strategic planning can also be used as a response to the problems in the tourism industry.

The Meta-SWOT model runs in a program consisting of a title window and seven interconnected windows. Its purpose is to guide decision-makers in an integrated process from the initial stage of the brainstorm to creating a ranked list of strategic priorities. This tool enables unlimited reviews of inputs, as decision-makers change their assessment during a planning activity.

Meta-SWOT is based on resource-based theory (RBV). This theory states that unique resources and capabilities are the main contributing factor in continuous competitive advantage.

The method of the present research is descriptive–analytical, and its purpose is applied. The method of data collection and the analysis of information is documentation and surveying. The Meta-SWOT analytical technique has been used to formulate the development strategy and explain the goals, resources, capabilities and environmental

factors. Data collection has been performed several times using the opinions of 39 experts, such that, except for the objectives of the research, all stages of the research were identified and prioritized with the help of experts. The research findings show that Meta-SWOT can eliminate many of the shortcomings and inadequacies of SWOT by avoiding subjective decisions and using a systematic and high-precision technique. On the other hand, this model tries to show that sustainable development and competition can be operational when tourism areas make the best use of internal resources and capabilities by rejecting the exogenous development perspective. Meta-SWOT can identify the position of Kaleybar and its villages in comparison with competitors in different dimensions, determine the value, scarcity, imitation and irreplaceability of resources and capabilities, examine the impacts of internal factors on controlling threats or the optimal use of external opportunities, and assess the appropriateness of using goals, resources and capabilities to play a constructive role in the optimization of the competitive advantages of sustainable tourism in rural areas. In this way, it can enable development and prosperity in the border city of Kaleybar and its villages.

Based on this, the studies and research conducted in this article show that Jolfa county is the biggest rival in the region to Kaleybar in terms of tourism capacities, with a weight of 4.26. Bazargan, with a weight of 4.11, Pars Abad, with a weight of 4, Maku with a weight of 3.55, Khoda afarin with a weight of 3.44 and Germi with a weight of 3.26 are next in the rankings. Additionally, regarding the potentials of the border areas of Jolfa, in terms of border tourism, it is considered the biggest rival in the region to Kaleybar, with a weight of 3.19. Khoda afarin, with a weight of 2.19, Bazargan with a weight of 2.16, Pars Abad with a weight of 2.14, Maku with a weight of 2.03 and Germi with a weight of 1.79 are next in the ranks. The resources and capabilities of Kaleybar were evaluated according to the theory based on resources and macro-environmental factors. Finally, the strategic appropriateness of each development factor and appropriate strategies for the development of the role and position of Kaleybar county were extracted. After examining the conditions having recognized and determined the capabilities and capacities of Kaleybar as well as its regional rivals, based on the research findings, the authors came to the conclusion that the abundant natural attractions are the most important advantage of Kaleybar. In addition, as this city has parks and coastal sidewalks and the possibility of attracting large internal and foreign investment in the region, it has the highest strategic fit. On the other hand, the component of political and governmental factors in attracting foreign tourists has the highest effective power, and the component of paying attention to integrated management in the field of tourism in the country has the highest degree of urgency.

## 6. Benefit from This Research and Suggestions for Other Research

Strategic planning is used as a tool to better perform activities and achieve the desired goals in the future. Strategic planning can also be used as a response to the problems in the tourism industry. Meta-SWOT is based on resource-based theory (RBV). This theory states that unique resources and capabilities are the main contributing factor to a continuous competitive advantage. Meta-SWOT can identify the position of Kaleybar and its villages in comparison with competitors in different dimensions, determine the value, scarcity, imitation and irreplaceability of resources and capabilities, examine the impact of internal factors on controlling threats or optimally using external opportunities, and assess the appropriateness of using goals, resources and capabilities in the optimal use of the competitive advantages of sustainable tourism in rural areas. In this way, it can enable development and prosperity in the border city of Kaleybar and its villages.

The findings and results of this study can be used with the help of politicians to develop tourism in border cities, especially the border city of Kaleybar and its villages, which can benefit from its potential for increasing welfare and development, and also establish relations with neighboring countries. Attracting tourist travelers contributes to the economic prosperity of the city, since tourism activity or the tourism industry can be a cultural, social and economic activity, such that it can maintain sustainable development

along with the development of sustainable tourism without harming nature. This research also had the goal of developing sustainable tourism in the border city of Kaleybar. In addition to economic, social and cultural activities, it is also possible to obtain relations with different neighboring countries, and to finance activities and jobs along with it. It can help to develop different types of border tourism in Iran, Azerbaijan and Armenia, for example that related to the use of medical services for the development of medical tourism (many annual travelers come to the eastern Azerbaijan province and Ardabil from the neighboring countries of Azerbaijan and Armenia to benefit from medical services) that can develop other types of tourism as well. Therefore, different and diverse fields related to the development of border tourism in Kaleybar city, the urban Azerbaijan province, as well as Ardabil and West Azerbaijan, can be found in these fields due to their proximity, language, cultural and climatic similarities, celebrations and tastes, as well as their sharing borders. Therefore, Kaleybar city, along with the surrounding cities, can be considered as a gateway for border tourism, and also a gateway for medical, cultural, social and religious tourism in various fields. We recommend the use of this knowledge and this article to researchers, as it can be used as a guide and tool for further research.

**Author Contributions:** Conceptualization, A.S.M.; formal analysis, A.S.M. and E.F.; investigation, A.S.M. and E.F.; methodology, A.S.M.; project administration, B.M.; resources, A.S.M. and L.S.; software, A.S.M. and E.F.; supervision, B.M.; validation, B.M.; visualization, A.S.M.; writing—original draft, A.S.M.; writing—review & editing, A.S.M. and L.S. All authors have read and agreed to the published version of the manuscript.

**Funding:** This research received no external funding.

**Institutional Review Board Statement:** Not applicable.

**Informed Consent Statement:** Informed consent was obtained from all participants involved in the study.

**Data Availability Statement:** Data sharing not applicable.

**Conflicts of Interest:** The authors declare no conflict of interest.

## Appendix A. Sample Tourist Photos of Kaleybar County

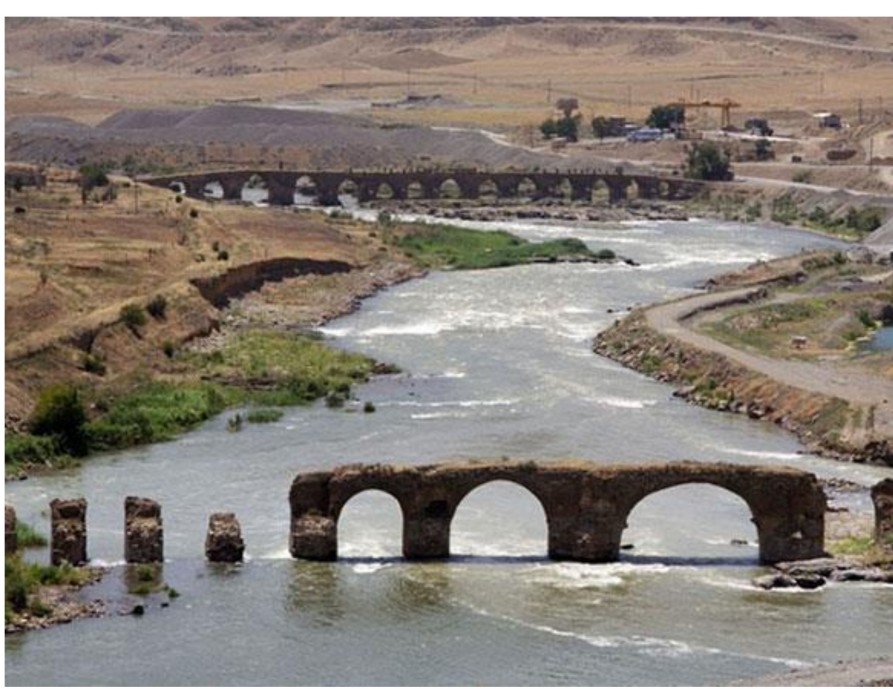

**Figure A1.** Khodaafarin historical bridge between Iran and Azerbaijan [by author].

The Azerbaijan Khodaafarin historical bridge between Iran and Azerbaijan on the river of aras. 1027 AD (first bridge), The bridges are located at a distance of 800 m from each other. A 15-span bridge, which was built in the 12th century, is in working order; the second, an 11-span bridge was built in the 13th century.

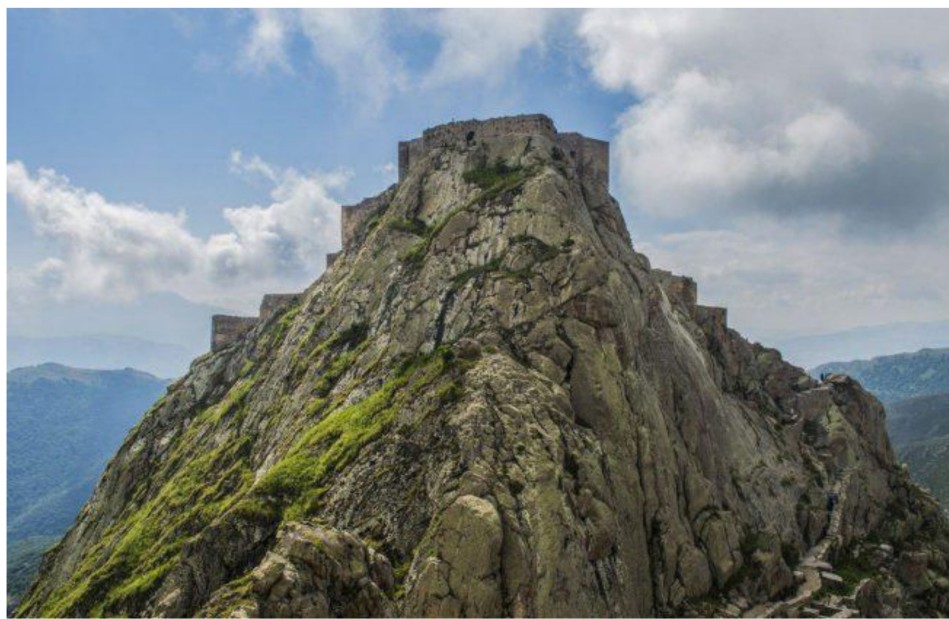

**Figure A2.** Babak Khorramdin historical castle [by author].

ABAK Fort (Ghaleh Babak or Babak Castle) located 5 km. southwest of Kalybar, at 2600 m. above sea level in East Azerbaijan Province city Kaleybar of Iran.

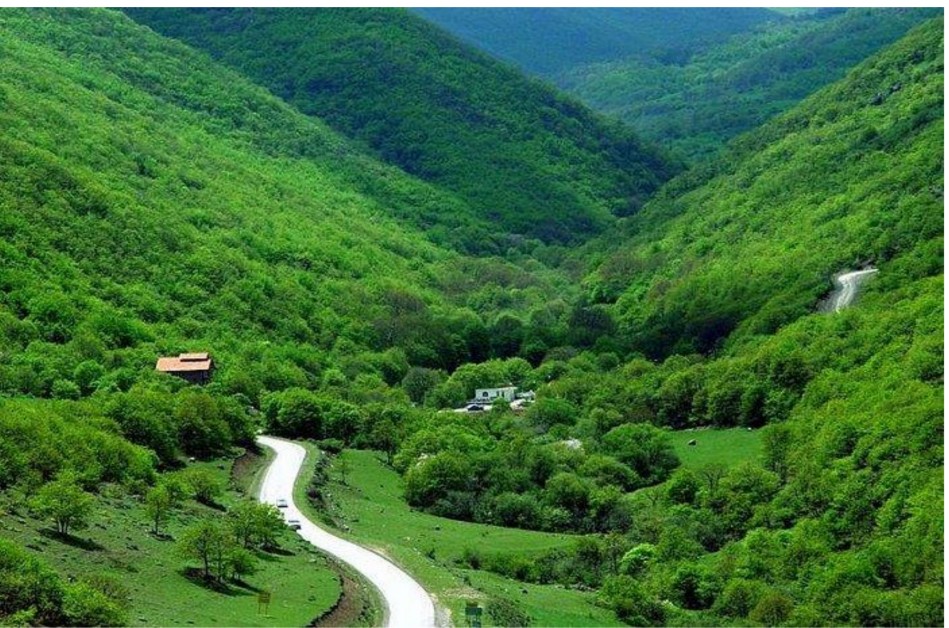

**Figure A3.** Arasbaran forest in Kaleybar [by author].

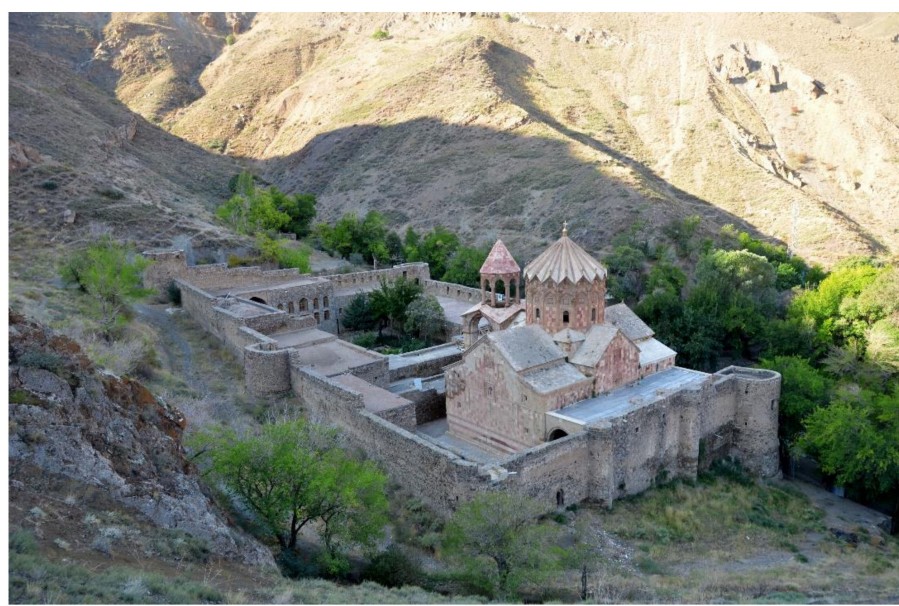

**Figure A4.** The Church of the Saint Stepanos [by author].

The Church of the Saint Stepanos Monastery (Built in the 9th century AD) is located 3 km south of the Aras River near the border with the Republic of Azerbaijan.

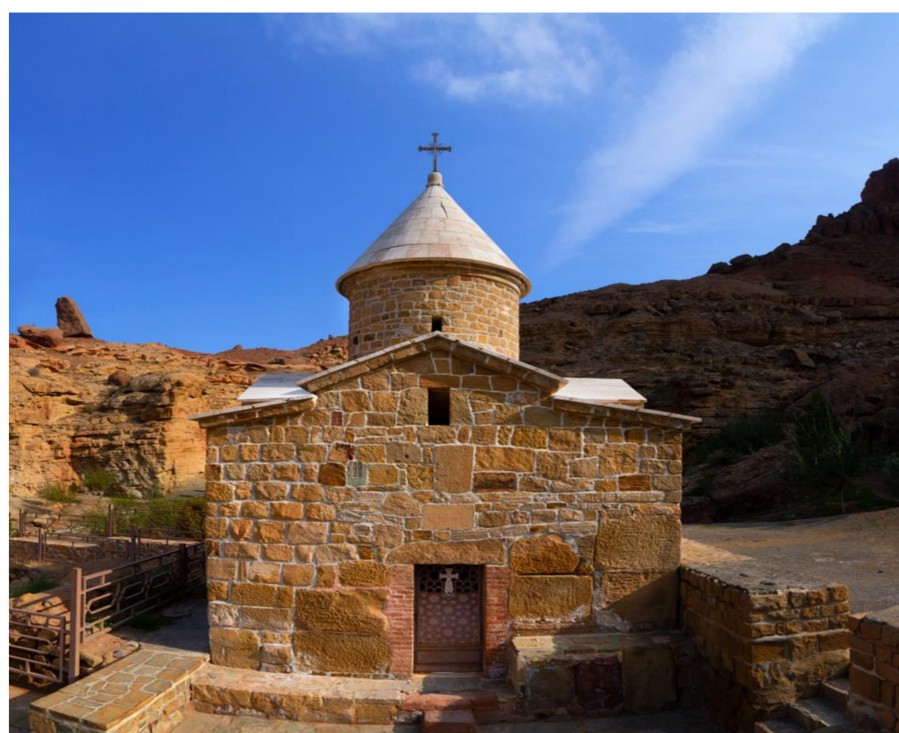

**Figure A5.** The Chapel of Chupan. The Chapel of Chupan, near the Aras River in East Azerbaijan Province, Iran [by author].

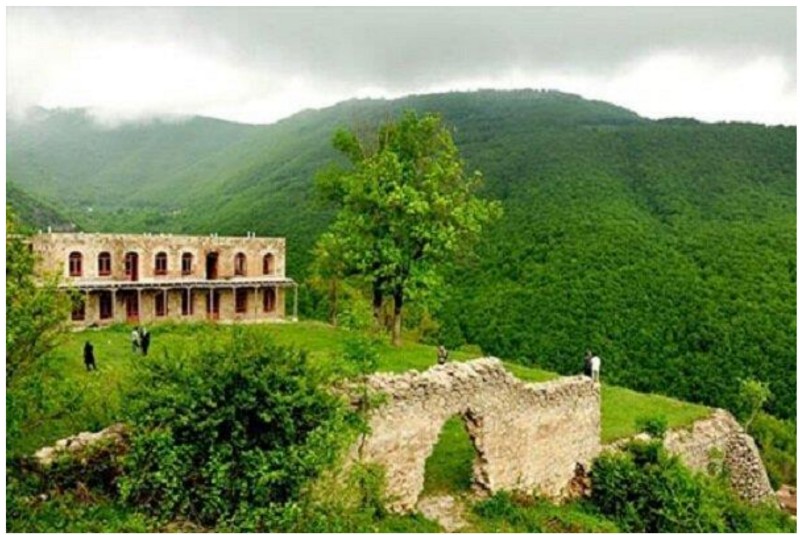

**Figure A6.** Ainalo Mansion, Arasbaran, Kaleybar [by author].

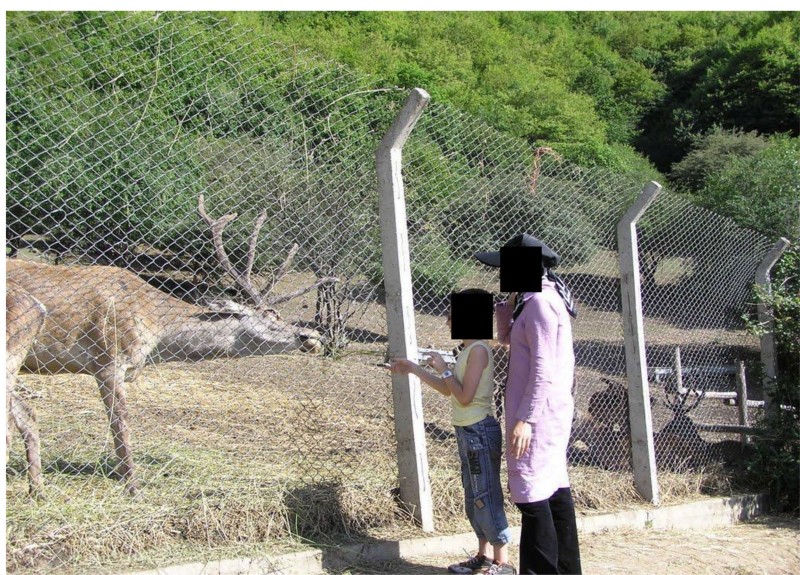

**Figure A7.** Kaleybar wildlife, Ainalo Arasbaran, Protected Area [by author].

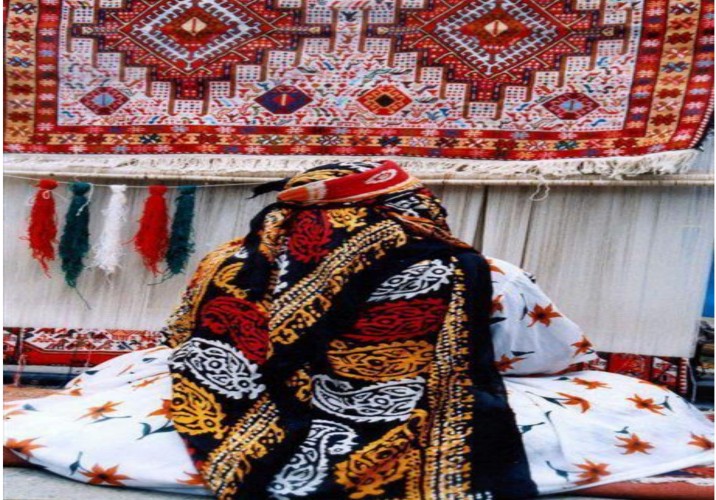

**Figure A8.** Kaleybar, Traditional carpet weaving [by author].

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
