# Peer review of "Border Tourism Development Strategies in Kaleybar Compared to Regional Rivals"

_sustainability, doi:10.3390/su132011400_

Round 1

Reviewer 1 Report

The topic is very interested and actual, but in this form it has a limited weight of scientific work.

Authors should improve the theoretical background. The Theoretical background in its current form is very weak, the main focus is on describing general facts about tourism. Authors should focus on researching determinants (resources, capabilities) in other tourist developing countries in the region that will serve as a basis for developing their own variables included in the metaswot analysis.

The strengths and opportunities are in the literature review well described, but weaknesses and threats should be covered as well.

Rows 72 – 86 and 95-108 are duplicated.

In the methodology section authors mentioned „experts„ but it is not described who the experts were, how they were selected and for what purpose.

Authors point out that „effectiveness was measured based on the percentage of impact they could have on achieving the research goals (table 2) “, but it is not clear how average scores are calculated, and how many variables was in total.

Discussion section is missing - authors should explain how their results extend the existing literature? Explain the importance and relevance of your results, possible improvements that can be made in order to further develop the concerns of your research, how your results reflect to your research questions etc.

In conclusion please add limitation of your research and contribution of the paper?

Author Response

Author's Reply to the Review Report (Reviewer 1)

Dear Reviewer,

Dear Professor,

Thank you for your comments

We have used your valuable comments and improved the article. Thank you very much. Thank you, we have modified everything that was said to correct the article and it has been highlighted in yellow on the article.

We have corrected all the said cases and if there is another point in your opinion, I will definitely do it.

The topic is very interested and actual, but in this form it has a limited weight of scientific work.

Thank you very much.  But We made it this way because it is subject-oriented and specific. If you suggest, we will definitely correct it. Please let us know if you have any comments.

Authors should improve the theoretical background. The Theoretical background in its current form is very weak, the main focus is on describing general facts about tourism. Authors should focus on researching determinants (resources, capabilities) in other tourist developing countries in the region that will serve as a basis for developing their own variables included in the metaswot analysis.

The theoretical foundations and sources mentioned by the Reviewers were read and added, and we reduced them due to the shortness of the article and the large volume. If your opinion is needed again, there are many theoretical bases, we will definitely do it.

The strengths and opportunities are in the literature review well described, but weaknesses and threats should be covered as well.

Advantages and disadvantages were mentioned in the article and according to the positive approach to tourism development in the city and also according to the sustainable tourism approach of the border city of Kalibar, these problems and disadvantages of tourism will be eliminated, so consider these cases in the article and do We have given.

Rows 72 – 86 and 95-108 are duplicated.

Was corrected.

In the methodology section authors mentioned „experts„ but it is not described who the experts were, how they were selected and for what purpose.

Thanks and the said items were corrected. It was mentioned in the methodology and also the number of specialists and the purpose of the research.

Authors point out that „effectiveness was measured based on the percentage of impact they could have on achieving the research goals (table 2) “, but it is not clear how average scores are calculated, and how many variables was in total.

The average and percentage of them are mentioned and also if you say it is another form, we would do it if it was possible to correct it.

Discussion section is missing - authors should explain how their results extend the existing literature? Explain the importance and relevance of your results, possible improvements that can be made in order to further develop the concerns of your research, how your results reflect to your research questions etc.

The section was added in the title and the results also explained the thematic relationship as well as the close results in innovation in the local dimension and also in the issue of sustainable tourism of border cities as one of the thematic innovations is consistent with the results and literature and other research. Benefits as well as future research and suggestions were discussed.

In conclusion please add limitation of your research and contribution of the paper?

We will definitely add these cases and the limitations of the research are the lack of data and the statistical information is accurate and up to date. The lack of a suitable statistical source and the difficulty of collecting this information, on the other hand, is due to the fact that this article is limited to mentioning many points in the Venice process and in the results, as well as in the theoretical foundations that can be published in a book. Due to the short length of the article, as well as the characteristics and restrictions of publishing and publishing, many titles, tables, maps, photos and diagrams have been omitted.

Also, the map and diagrams were drawn again, but this is a problem because when we copy an image and paste it into a Word file, the quality of the image and diagram map decreases. To solve this problem, if you want, I can separate the files and images as well as the diagrams as the output of GIS software, as well as the desired Meta-SWOT t model, which does not output the image itself, and we take photos from it, so when We put it in Word, the quality is a bit low, and finally, we tried to make it high quality. The map was prepared with 500 DPi.

Thank you, we have modified everything that was said to correct the article and it has been highlighted in yellow on the article.

Finally, thank you and your comments and views were very valuable and useful to us.

Sincerely, Amin

Reviewer 2 Report

The paper is generally well written and structured. The literature review is pertinent. The objectives have been well explained and the methodology is clear.

Please consider the following comments for the revision of your manuscript:

  • Study area: It may be interesting to include more data/information about the tourism offer and demand in the study area presentation.
  • Review "Strategic Planning in the Tourism Industry" because some sentences are repetitive. Insert references for “the tourism industry can be one of the most profitable industries in the world 151 from 2000 to 2020 and even….”.
  • In-text-references: it is important to verify references format. For example:

    Line 45: missing the year of “(Pour Ahmad, et. al)”

    Line 46: change “(Madhoushi, 1382: 26)” with “(Madhoushi, 2003: 26)”

    Line 51: change “(Zarei et al, 1391, 75)” with “(Zarei et al, 2012: 75)”

    Line 81: insert comma and erase point “Land Management and Planning Organization, 2016”

    Line 85 and 108: insert comma “UNESCO MAB, 2010”

    Lines 103-104: insert semicolon after “Ministry Country, 2016”

    Line 128: change “(Eaagles et all, 2002: 25)” with “Eaagles et al, 2002: 25)”

    Line 136: change “(Rastaghalam Et al., 2010:124)” with “(Rastaghalam et al, 2010:124)”

    Line 161: change “(Almasi et al., 2011, p. 98)” with “(Almasi et al, 2011: 98)”

    Line 167: change “(Shields & Hughes, 2006: 20)” with “(Shields and Hughes, 2006: 20)”

    Line 176: change “(Titus et all, 2012: 14)” with “(Titus et al, 2012: 14)”

    Line 188: change “(Mowforth & Munt, 188 2003: 240)” with “(Mowforth and Munt, 188 2003: 240)”

    Line 200: change “(Lorio & Wall, 2012: 1440)” with “(Iorio and Wall, 2012: 1440)”

    Line 212: missing the year of UNESCO’s publication

    Line 250: change “(Agarwal ET al, 2012:13)” with “(Agarwal et al, 2012:13)”

    Lines 346-347: change “(Jones and Hill, 2013, 335)” with “(Jones and Hill, 2013: 335)”

  • References list:

    It is important to verify references format and also check numbers: 19, 20, 25, 26, 36, 37.

  • Specific comments

    Line 36: add a space after the purposes and erase space after the parenthesis

    Line 69: erase “studying”

    Lines 76-77: check

    Line 141: Tourism is one of the most important sub-sectors of the tourism industry ??????

    Lines 281-282: check

    Line 298: erase endpoint after Kaleybar

    Line 306: insert the endpoint

    Line 371: insert the endpoint

    Figures:

    Figure 1: check. Some words are incomplete

    Figure 2: This figure is difficult to understand. Improvements need to be made. 

    Figure 4: check the typo in the figure

Author Response

Author's Reply to the Review Report (Reviewer 2)

Dear Reviewer,

Dear Professor,

Thank you for your comments

We have used your valuable comments and improved the article. Thank you very much. Thank you, we have modified everything that was said to correct the article and it has been highlighted in yellow on the article.

We have corrected all the said cases and if there is another point in your opinion, I will definitely do it.

Please consider the following comments for the revision of your manuscript: (Thank you, we have modified)

Study area: It may be interesting to include more data/information about the tourism offer and demand in the study area presentation. (Thank you, we have modified)

Review "Strategic Planning in the Tourism Industry" because some sentences are repetitive. Insert references for “the tourism industry can be one of the most profitable industries in the world 151 from 2000 to 2020 and even….”. (Thank you, we have modified)

  • In-text-references: it is important to verify references format. For example:

Line 45: missing the year of “(Pour Ahmad, et. al)” (Thank you, we have modified)

Line 46: change “(Madhoushi, 1382: 26)” with “(Madhoushi, 2003: 26)” (Thank you, we have modified)

Line 51: change “(Zarei et al, 1391, 75)” with “(Zarei et al, 2012: 75)” (Thank you, we have modified)

Line 81: insert comma and erase point “Land Management and Planning Organization, 2016” (Thank you, we have modified)

Line 85 and 108: insert comma “UNESCO MAB, 2010” (Thank you, we have modified)

Lines 103-104: insert semicolon after “Ministry Country, 2016” (Thank you, we have modified)

Line 128: change “(Eaagles et all, 2002: 25)” with “Eaagles et al, 2002: 25)” (Thank you, we have modified)

Line 136: change “(Rastaghalam Et al., 2010:124)” with “(Rastaghalam et al, 2010:124)” (Thank you, we have modified)

Line 161: change “(Almasi et al., 2011, p. 98)” with “(Almasi et al, 2011: 98)” (Thank you, we have modified)

Line 167: change “(Shields & Hughes, 2006: 20)” with “(Shields and Hughes, 2006: 20)” (Thank you, we have modified)

Line 176: change “(Titus et all, 2012: 14)” with “(Titus et al, 2012: 14)” (Thank you, we have modified)

Line 188: change “(Mowforth & Munt, 188 2003: 240)” with “(Mowforth and Munt, 188 2003: 240)” (Thank you, we have modified)

Line 200: change “(Lorio & Wall, 2012: 1440)” with “(Iorio and Wall, 2012: 1440)” (Thank you, we have modified)

Line 212: missing the year of UNESCO’s publication (Thank you, we have modified)

Line 250: change “(Agarwal ET al, 2012:13)” with “(Agarwal et al, 2012:13)” (Thank you, we have modified)

Lines 346-347: change “(Jones and Hill, 2013, 335)” with “(Jones and Hill, 2013: 335)” (Thank you, we have modified)

  • References list:

It is important to verify references format and also check numbers: 19, 20, 25, 26, 36, 37. (Thank you, we have modified)

  • Specific comments

Line 36: add a space after the purposes and erase space after the parenthesis (Thank you, we have modified)

Line 69: erase “studying” (Thank you, we have modified)

Lines 76-77: check (Thank you, we have modified)

Line 141: Tourism is one of the most important sub-sectors of the tourism industry ?????? (Thank you, we have modified)

Lines 281-282: check (Thank you, we have modified)

Line 298: erase endpoint after Kaleybar (Thank you, we have modified)

Line 306: insert the endpoint (Thank you, we have modified)

Line 371: insert the endpoint(Thank you, we have modified)

Figures:

Figure 1: check. Some words are incomplete(Thank you, we have modified)

Figure 2: This figure is difficult to understand. Improvements need to be made. (Thank you, we have modified)

Figure 4: check the typo in the figure (Thank you, we have modified)

Also, the map and diagrams were drawn again, but this is a problem because when we copy an image and paste it into a Word file, the quality of the image and diagram map decreases. To solve this problem, if you want, I can separate the files and images as well as the diagrams as the output of GIS software, as well as the desired Meta-SWOT t model, which does not output the image itself, and we take photos from it, so when We put it in Word, the quality is a bit low, and finally, we tried to make it high quality. The map was prepared with 500 DPi.

Thank you, we have modified everything that was said to correct the article and it has been highlighted in yellow on the article. We have corrected all the said cases and if there is another point in your opinion, I will definitely do it.

Finally, thank you and your comments and views were very valuable and useful to us.

Sincerely, Amin

Reviewer 3 Report

The manuscript with the title Border tourism development strategies in the Kaleybar Compared to regional rivals can be potentially published after thorough revision and resubmitted again.

Author Response

Author's Reply to the Review Report (Reviewer 3)

Dear Reviewer,

Dear Professor,

Thank you for your comments

We have used your valuable comments and improved the article. Thank you very much. Thank you, we have modified everything that was said to correct the article and it has been highlighted in yellow on the article.

We have corrected all the said cases and if there is another point in your opinion, I will definitely do it.

Comments and Suggestions for Authors

The manuscript with the title Border tourism development strategies in the Kaleybar Compared to regional rivals can be potentially published after thorough revision and resubmitted again.

Dear Professor

Thank you for your comments and kind comments

Dear Reviewer, Thank you very much for saying good things.

 We have corrected all the things you said in the PDF file(peer-review-14329305.v2.pdf)  and we have specified them in the article.

Also, the map and diagrams were drawn again, but this is a problem because when we copy an image and paste it into a Word file, the quality of the image and diagram map decreases. To solve this problem, if you want, I can separate the files and images as well as the diagrams as the output of GIS software, as well as the desired Meta-SWOT t model, which does not output the image itself, and we take photos from it, so when We put it in Word, the quality is a bit low, and finally, we tried to make it high quality. The map was prepared with 500 DPi.

Thank you, we have modified everything that was said to correct the article and it has been highlighted in yellow on the article.

Finally, thank you and your comments and views were very valuable and useful to us.

Sincerely, Amin

Round 2

Reviewer 1 Report

No additional comments. The Authors improved the paper according to instructions.

Reviewer 3 Report

The authors done all of necessary things. This paper now present really great work. For my opinion this manuscript can be accepted. 

Sincerely, the Reviewer#